# Attention Paid by Children of Rural Mapuche, Urban Mapuche and Non-Indigenous Chilean Backgrounds to Interactions Directed at Others

**DOI:** 10.3390/bs14080689

**Published:** 2024-08-08

**Authors:** Rebeca Muñoz, Paula Alonqueo

**Affiliations:** 1Departamento de Ciencias del Lenguaje y Literatura, Facultad de Educación, Universidad Católica de la Santísima Concepción, Concepción 4030000, Chile; 2Departamento de Psicología, Facultad de Educación, Ciencias Sociales y Humanidade, Universidad de La Frontera, Temuco 4811230, Chile

**Keywords:** third-party attention, LOPI, indigenous, learning

## Abstract

This study aimed to establish differences in third-party attention through a toy-building activity among children between 9 and 11 years old from three cultural backgrounds: Rural Mapuche, Urban Mapuche and non-Indigenous Chilean. It was also examined whether third-party attention is related to learning a previously observed activity. Third-party attention involves maintaining two or more foci of interest simultaneously without losing attention and or interrupting the course of a task. It is of interest to study because it may undergo changes as a result of exposure to schooling. Given that these groups differ in cultural practices and years of formal schooling, the hypothesis was that it might be possible to identify differences in their attention patterns. The results showed that it seems like practices of Rural Mapuche families encourage third-party attention much more so than the other groups; therefore, the learning of skills arises in constellations of cultural practices that involve children’s living conditions and guide their development.

## 1. Learning by Observing in Indigenous American Communities

Attention to events that are not directed at oneself—attention to others—has been described as a learning strategy frequently used by indigenous communities of the Americas [1]. In fact, ethnographic research conducted in these communities has shown that children are integrated into a wide range of community and family activities and are encouraged to attentively observe the events around them [2,3,4,5].

Research has shown that by observing, listening and paying attention to ongoing events, children in these communities acquire skills to progressively contribute to family activities and community endeavors [6,7,8]. This way of learning, addressed in the literature as Learning by Observing and Pitching In (LOPI) [9], has been integrated into the routine and cultural values of the Americas’ indigenous communities due to the importance of natural real-world tasks [4]. This occurs from children’s desires to cooperate and undertake actual active participation in the activities [8,9].

According to LOPI, in these communities, observation and attentive listening are a strategic component of childcare that is manifested through commitments to help the community and the family [8,10]. These imply responsibility and trust in each member of the community by sharing collective efforts for children to increase their participation in activities in a socio-affective framework that promotes acute and intense attention [11].

The broad and keen observation capabilities developed by people in indigenous communities, or in communities of the Americas with indigenous heritage, can manifest itself in different ways, such as intense and sustained attention to practices in order to be able to contribute when necessary; as simultaneous attention to multiple foci without detracting from one to the other and without interrupting or pausing the course of a task; or as third-party attention, that is, attention that focuses on interactions directed at other people in order to learn from what others do (not participating oneself, but as a peripheral observer) [1,3,12,13]. The latter type of attention is of interest for this study.

## 2. Cultural Patterns in Attending to Information Directed to Others

Research conducted within the fifth facet of the LOPI model framework shows that attention to situations directed at others (third-party attention) is a learning strategy widely used by indigenous communities, even by people with indigenous origins with extensive formal schooling [9,14].

Thus, when children are included in community and family tasks, they can collaborate in advance, give attentive suggestions and avoid distractions that interrupt the course of activities [10,15]. They can take the initiative to manage their own attention, adjust to the pace of collaboration, use nonverbal cues and execute a task without the need to receive positive feedback from adults [7,16].

On the other hand, in Euro-American families with extensive formal schooling (12 or more years) it is less frequent for children to participate in family and community helping activities, as caregivers have skilled jobs outside the home. Children’s practices focus on school-related activities, and, therefore, it is considered inappropriate to give children responsibilities other than school-related ones [5,17,18]. Caregivers and parents of these children organize daily activities that demand their involvement through lessons; for example: after school, they eat a snack, go to piano lessons and then do homework [2,7]. For this reason, opportunities to learn by attending to situations that are not directed to them are scarce; these children have fewer opportunities to develop social responsibility, observation and collaboration [2,19].

A study by Correa-Chávez and Rogoff [20] explored third-party attention in children from three communities: traditional Mayan families with limited formal schooling (2 years on average), Kaxlaan Mayan families with extensive formal schooling (12 years on average) and middle-class Euro-American families with high rates of formal schooling (16.8 years on average). A protocolized turn-based toy construction task performed in pairs of siblings was implemented; one of the children did not participate directly but could observe while waiting for their turn. A week later, the children were individually and unexpectedly invited to assemble the toy they had had the opportunity to observe the assembly process of. The children from traditional Mayan families spent much of their time attending to the construction of the toy and needed less help to assemble it a week later, whereas in the Euro-American sibling groups and the Kaxlaan Mayan sibling groups, less attention had been paid, and the help required was greater.

Similar studies [12,21] involved children from communities with Mesoamerican indigenous heritage (immigrants to the U.S.) from families with limited formal schooling (on average 6 years) and children from families with extensive formal schooling (12 years or more). Children with indigenous heritage whose families had limited formal schooling were especially prone to observe and listen to the other child’s construction of the toy, even when the interactions were not directed at them. For this reason, seven days later, they needed less help to assemble the toy whose construction they had observed earlier. On the contrary, children (12 or more years of age) whose mothers had more formal schooling showed less third-party attention and, therefore, needed more help to remember how to assemble the toy. These results show that third-party attention facilitates the learning of an activity performed by others [1].

It is of interest to study third-party attention because it may undergo changes as a result of exposure to formal schooling. In school, teaching is predominantly separated from productive activities; it is individually oriented and direct, it encourages selective attention, and teachers directly control students’ behavior [9]. In the case of children with indigenous heritage, they must share cognitive resources to maintain third-party attention to surrounding events while keeping the selective individual attention required in school [18,22].

## 3. Differences from the Previous Studies

This research was a replication of Correa-Chávez and Rogoff [20] study of third-party attention in children from three cultural groups: rural Mapuche indigenous, urban Mapuche indigenous (or with indigenous heritage) and urban non-indigenous backgrounds. It aimed to provide evidence of attention to interactions directed to others within these groups, which differ in their cultural practices, and the extent of formal schooling; because of this, it was considered possible that they might also differ in their attention patterns [23,24]. This study had not been undertaken for these groups before, and, unlike the original work by Correa-Chávez and Rogoff [20], it was not applied to sibling dyads but to student friends.

The rural Mapuche group is characterized by maintenance of both the vernacular language and traditional Mapuche practices. The community has an agricultural–livestock economic system, and formal schooling levels correspond, on average, to 8.2 years in La Araucanía, Chile [25]. Therefore, it can be assumed that Mapuche cultural practices have not been strongly permeated by school-like practices. Parenting practices are shared in a system of multi-parenting in the extended family, and interaction with adults is horizontal and collaborative, as children are considered legitimate participants [26]. Learning occurs in collaboration with the group through observation of the daily tasks of the family and community [27]. Children develop the initiative to contribute and collaborate in tasks while simultaneously attending to the needs of the group [28].

On their end, the urban Mapuche group corresponds to families with a history of forced migration from rural to urban areas due to the loss of land or the search for specialized jobs in a free market economic system [29]. Currently, migration has continued due to the search for greater educational offerings and the absence or scarcity of secondary education establishments [30].

Some families have constant contact with their communities of origin in a relationship of socio-spiritual dependence between the countryside and the city [29]. They maintain the use of Mapudungun, practice cultivation of the land and participate in indigenous spiritual ceremonies in their communities of origin [31,32]. However, the daily routine of these families has adapted to school institutions and adult work outside the home, thereby decreasing child collaboration in domestic activities [33].

Finally, non-Mapuche families in urban areas have extensive levels of formal schooling (11.9 years) [25]. Frequently, these families have practices focused on school activities, and for this reason, children have little time available to collaborate in household activities [34].

Since the 1970s, non-indigenous Chilean children have stopped helping with household chores due to a change in parental beliefs, as it is considered desirable for them to study and graduate from high school [35]. Families privilege individualism, good performance in school activities and autonomy regarding the decisions and practices of each family member [36,37,38].

Family composition in urban areas tends to be nuclear [39]. Most have adult supervision and care and, therefore, do not have as many opportunities to observe and participate in group activities that enhance collective awareness of action [39]. All this contributes to the construction of an independent self in children [40].

Based on the aforementioned information, it is assumed that Mapuche children living in rural areas, whose families participate in indigenous practices and have little formal schooling, use third-party attention more frequently in activities in which they do not participate directly. On the other hand, it is expected that urban Mapuche children and urban non-indigenous children, whose families have more formal schooling, use third-party attention less frequently.

This research aims to establish differences in third-party attention among rural Mapuche, urban Mapuche and urban non-Indigenous children. It also examines whether third-party attention paid to interaction is related to learning a previously observed activity.

## 4. Methodology

### 4.1. Participants

#### Study Population

In Chile, the Mapuche people are the largest indigenous group, with a total population of over one million inhabitants (1,745,147) [25], of which 471,742 are children and adolescents [41]. The Mapuche participants in this study resided in rural areas of La Araucanía, whereas the urban Mapuche and urban non-Indigenous children resided in the city of Temuco.

To access the participants, six schools were selected; four were urban and two were rural. The latter have a high percentage of indigenous enrollment, operate with a multigrade modality and have an Intercultural Bilingual Education program (in Spanish, EIB).

Purposive sampling was performed according to the following inclusion criteria: (a) cultural group (exclusively Mapuche and non-Mapuche Chilean children), determined by self-ascription; (b) urban or rural origin; and (c) age range from 9 to 11 years old. The exclusion criteria were (a) belonging to a cultural group other than Mapuche or non-Mapuche; (b) being outside the age range; and (c) having a developmental disorder, such as cognitive or motor disability. No groups of immigrant children (e.g., Venezuelan or Haitian) or any group of other indigenous children different from Mapuche were included.

The sample consisted of 102 children between 9 and 11 years of age (*M* = 9.9)—58 girls and 44 boys. Participants were grouped in dyads, one year apart in age, within the same cultural group: 19 rural Mapuche (20 girls and 18 boys), 13 urban Mapuche (15 girls and 11 boys) and 19 urban non-Mapuche (23 girls and 15 boys). The participating children were given a sociodemographic questionnaire with questions on age, cultural group, sex, origin, level of formal schooling, type of parental work and participation in Mapuche ceremonies.

Since formal schooling is one of the many variables that could affect the way in which children pay attention and learn—for example, a large part of the experience in formal schooling may be related to practices that encourage the development of individual and selective attention [15,21]—the parents’ level of formal schooling was also recorded as an emerging variable.

The mothers of rural Mapuche children had an average of 9.9 years of formal schooling, and the fathers had 9.4 years. Most of the mothers were housewives, and the fathers were agricultural workers. A total of 68.4% of the rural Mapuche children participated in spiritual ceremonies with their family and community, 18.4% participated only in Mapuche activities held in their schools and 13.2% did not participate in any indigenous ceremonies.

The average formal schooling of the mothers of the urban Mapuche children was 11.4 years, and the average formal schooling of the fathers was 11.5 years. Most of the mothers were housewives, and the fathers were employees in the private sector whose workplaces were outside the home. In total, 23.1% of the urban Mapuche children participated in Mapuche ceremonies with their family and/or community; 26.9% participated only in ceremonies organized by their school; and 50% did not participate in any Mapuche ceremonies at all.

Mothers of urban non-Mapuche children had an average of 12.8 years of formal schooling, whereas fathers had 12.1 years of formal schooling. Half of the mothers were housewives, and the fathers were employed in the private sector.

### 4.2. Instruments

#### Third-Party Attention

According to Ref. Correa-Chávez and Rogoff [20], the third-party attention task assesses attention directed to third-party interactions and learning in a toy-building activity. This task has been used for North American children and indigenous children from the Americas [12,21].

The application of the task was performed in two sessions. The first session assessed third-party attention and was developed with a dyad of children from the same cultural group. Toy Lady (research assistant) instructed each child how to assemble a toy, one at the time, taking turns. The older child was shown how to assemble a foam mouse while the younger child waited at an adjacent table and played with a distractor device (called a do-nothing machine). Once the first child finished making the toy, the roles were exchanged; she showed the younger child how to assemble a different toy (an origami frog) while the first child (the oldest) waited and played with the distractor device. In the second session, with a one-week lag, each child, individually, returned to collect the toy they had assembled earlier. Toy Lady invited them to assemble the toy that their partner assembled the week before. This invitation was unexpected, since neither child knew that there was a second session.

It should be mentioned that Toy Lady was an undergraduate student, and she was not Mapuche but had knowledge of the cultural practices of this community. She was trained in the application script to limit the directions and not to direct the attention of the child not involved in the construction of the toy [20]. Also, none of the sessions were conducted in Mapudungun, as most of the children were native Spanish speakers.

### 4.3. Procedure

#### 4.3.1. Session 1: Observing the Construction of the Toy

Each dyad was taken by Toy Lady to a room adapted with pictures and objects so that the children felt it was recreational and not a classroom. The room had two tables, three chairs and a camera for videotaping (Figure 1). The position of the uninvolved child’s chair and table was placed so that he or she was not directly involved in the toy’s construction but could turn to watch the toy construction activity. Toy Lady directed the toy construction activity following the indications of an ad hoc manual [20].

The models of the two toys to be assembled were on the main table. Toy Lady introduced herself and engaged the children in small talk, for example, about what they had done over the weekend, so they were more comfortable for the activity. In addition, she told them that building the toys was not graded and that it was a recreational activity. Toy Lady asked the children if they knew the toys. If any child expressed that he or she knew them and knew how to assemble them, that video recording was discarded from the analysis sample. She then took the younger child to the side seat and explained that she would start working with their older partner who was going to assemble a mouse; next, Toy Lady specified that the younger one would later work on making a frog. Thereafter, she handed the uninvolved child the distractor toy to play with (the do-nothing machine). The distractor toy was a wooden block with a crank that the child could turn [20].

Next, Toy Lady sat down with the older child at the head table and placed two sets of materials to assemble the toy mouse in front of them. She pointed out that each had their own set so that the involved child could watch her for guidance.

Once the mouse construction was complete, Toy Lady asked the older child to exchange places with their partner, receiving the do-nothing machine, thus switching roles so that the younger child could start working on an origami frog with Toy Lady.

At the end of the session, Toy Lady asked the children to write their names on a paper bag to keep their creations. She told the dyads that their toys would be put away for a few days until others had the chance to assemble their own. Regardless, she let them know they could pick up their toys the following week.

#### 4.3.2. Session 2: Learning of the Uninvolved Child

A week later, each child, individually, returned to the room to receive the toy they assembled earlier. This time, Toy Lady gave them the opportunity to assemble the toy their partner had assembled, saying that she had extra materials she thought the child might like to use. Toy Lady put the supplies the child needed to work on the table; next, she mentioned she had tasks to attend to but, however, that she was available to help if needed. Subsequently, she picked up a book and began to read, seemingly engrossed. Her task was to discreetly supervise the child’s construction of the toy, following the instructions in the manual that contained a standard set of clues for the elaboration of the toys—four steps for the mouse, five steps for the frog [20].

At each step, Toy Lady waited approximately 10 s to allow the child to begin. If they did not complete the step spontaneously, Toy Lady would give them a hint specifying which part to work on (e.g., “the first thing is the body”) or which material to start with (e.g., “try to start with the yarn”). If the child continued to show difficulty through the process, Toy Lady would give them a more specific hint about what to do with the materials. If the child needed more help, Toy Lady demonstrated the step, and finally, if the child could not do the step, she would complete it and return to her reading, telling the child to proceed. Upon completion of the toy construction, the child took their mouse and frog with them. All participants in the same school completed the first session before starting the second one.

### 4.4. Ethical Safeguards

This research was presented to the school administrators by means of a document explaining the study. After obtaining the principal’s signed authorization for the administration of this study, it was agreed with the head teachers of each class to deliver the informed consent forms to the children’s caregivers. Upon receipt of these signed consents, and prior to the administration of third-party attention tasks, the authorized children were asked to sign their informed assent, which was explained by Toy Lady. The schedule to apply the instruments was set in agreement with the teachers, and they were also asked for information about the children to form the dyads for the third-party attention tasks. The teachers indicated the children who were friends with each other or with whom there were no good relations. They also corroborated information about the indigenous ancestry of the children. The dyads were formed by children who were friends, who were at least one year apart and who were part of the same cultural group.

In addition, sociodemographic data of the children were requested. The data were collected from both the student information sheets that each school handled and the information provided by the main teachers. After the application of third-party attention tasks, the sociodemographic questionnaire information was completed with each child.

### 4.5. Coding

#### 4.5.1. Coding of Variables: Session 1

Two coders worked on the video recordings according to the coding manual and the list of variables elaborated by Correa-Chávez and Rogoff [20]. They were trained, did not know the objectives of the research and were unaware that the task had two sessions.

The first session was coded 100% by each assistant to ensure the validity of the data. The coding process followed the indications of the original study [20]; each video was divided into five-second segments, and in each segment, the predominant form of attention of the uninvolved child, who was of interest for this study, was evaluated. The coding considered the gestures of the target child, the direction of their body posture and their glances towards the toy construction process during each five-second segment [21]. The variables coded were sustained attention to the construction, casual glance and inattention, exclusive in the same segment; the scores for these were calculated according to the percentage frequency of occurrence of the segments in the whole sample. [Because the children’s performance varied in duration, the analysis segments varied from child to child. Therefore, the number of segments for each child served as the basis for calculating the frequency of each type of attention. For example, if a child had 21 segments, this would be his 100%, which was distributed as follows: 24% (5 segments) of sustained attention; 5% (1 segment) of casual glance; and 71% (15 segments) of inattention].

For all the variables in the first session, intercoder agreement was calculated using the Kappa coefficient, which demonstrated a substantial level of intercoder reliability with values between κ = 0.70 and 0.80, *p* < 0.01 [42].

The coding variables used were as follows:

Sustained attention: intense observation of toy construction during the majority of the segment. It involves active attention, i.e., an alert body posture and a fixed gaze on the toy-making activity.

Casual glance: brief and very sporadic interest in the toy construction, approximately 1 s within the segment. The uninvolved child’s glances do not maintain a focus on the activity. For the rest of the segment, he attends to something other than the making of the toy.

Inattention: lack of attention to toy making. This includes concentrating on playing on the do-nothing machine, looking at objects in the room or being entertained by something in the room, humming a song or rocking in the chair.

Only the group of rural Mapuche children did not interrupt the toy-building activity. As interruption and types of inattention were not part of the objectives of our study, this is not reported.

#### 4.5.2. Coding of Variables: Session 2

The variable of the second session, help, was coded 100% by the first coder, and the second coder coded 40% of the data. The intercoder reliability of the second session was calculated with the intraclass correlation coefficient by continuous scores and the proportion of total variance. Reliability was ICC = 0.83, *p* < 0.01, IC 95% [0.68, 0.92], indicating excellent intercoder reliability agreement [42].

Help needed: the number of progressive hints each child needed to complete each step of the construction of each toy (frog or mouse) was counted. The level of hints given to the child was obtained according to the points assigned to each step—0 if they completed the step without help from Toy Lady; 1 if they needed a small hint about the step or the materials to use; 2 if the child needed a bigger hint about what to do with the materials; 3 if Toy Lady showed the child a bit of the step; and 4 if Toy Lady completed the entire step.

The maximum number of the help score for the construction of the mouse was 16 points (the construction of this toy had four steps), and for the frog, it was 20 points (the construction of this toy had five steps). Since the maximum scores were different, this variable was transformed according to an equal proportion of help for the two toys. Consequently, the minimum score was 0 points, and the maximum score was 80 points, in either case.

### 4.6. Data Analysis

The data from the first session were analyzed using the time segments. Despite the fact that the children varied in how long they took to construct the toys, there were no statistically significant differences between the children (first or second) in the three cultural groups with respect to the time they took to assemble the toys in the first session—*F*(1, 100) = 0.00, *p* = 0.99 and η^2^ = 0.00. This result was used as rationale for determining that the first session variables should not be standardized to allow for parsimony, to avoid difficulties in interpretations of the results and to perform main effects analyses of the differences between groups [43].

A descriptive analysis of the dependent variables was performed (calculation of mean and standard deviation). ANOVA (univariate analysis of variance), with planned comparisons, was used to test predictions about sustained attention and help needed by the children in the three groups. This type of analysis was appropriate due to the directional hypotheses between groups [43]. The variables of casual glance and inattention were analyzed with ANOVA, and the Games–Howell statistic was applied in post hoc tests, since there were no previous directional predictions. Bivariate correlation analysis with Pearson’s *r* was used to investigate the relationship between the variables of sustained attention and learning.

The analyses of differences by sex were performed with a *t*-test for independent groups.

## 5. Results

### 5.1. Attention to the Toy-Building Activity (Session 1)

The children who observed first and those who observed second showed similar patterns of attention [*F*(2, 99) = 0.0, *p* =1.0], so the data from both children were combined to compare the three groups. Table 1 presents the means and standard deviations obtained in the first session variables.

The results of the one-way ANOVA showed a significant univariate main effect of sustained attention among the three groups of children—*F*(2, 99) = 3.52, *p* = 0.03, η^2^= 0.06, IC 95% [0, 69]. The results of the planned comparisons showed that the sustained attention of the rural Mapuche children was significantly higher than the attention of the children in the other two groups (*t*(54.43) = −2.39, *p* = 0.02). Specifically, rural Mapuche children presented a significantly higher amount of sustained attention both compared to non-Mapuche children (*t*(64.69) = −2.16, *p* = 0.03) and to their urban Mapuche peers (*t*(60.50) = −2.22, *p* = 0.03). There were no significant differences between urban Mapuche and non-Mapuche children with respect to sustained attention (*t*(58.77) = 0.10, *p* = 0.92).

By means of an ANOVA, it was found that the differences between the groups in the casual glance variable were statistically significant (*F*(2, 99) = 3.69, *p* = 0.03, η^2^ = 0.07, IC 95% [0, 29]). With Games–Howell, it was found that the casual glance in rural Mapuche children had a significantly lower amount of time compared to non-Mapuche children (*p* = 0.02). The other comparisons between groups were not statistically significant.

Regarding inattention, the three cultural groups showed similar scores, i.e., there were no significant differences in the amount of time they did not pay attention to the construction of the toy (*F*(2, 99) = 0.12, *p* = 0.88, IC 95% [10, 97]).

Finally, there were no statistically significant main effects between boys and girls on the dependent variables for the three groups (*p* > 0.05)—sustained attention (*p* = 0.73), casual glance *(p* = 0.66) and inattention (*p* = 0.69).

### 5.2. Learning: Amount of Help Needed in Toy Construction (Session 2)

In order to assess learning, planned comparisons were made with the amount of help needed by the children to assemble the toy, which was observed in session 1 but built by the other member of the dyad.

Table 2 presents the scores on the amount of help required and the correlation matrix between sustained attention (session 1) and help needed (session 2).

The amount of help needed to build the toy observed in session 1 did not differ in a statistically significant way among the three cultural groups (*F*(2, 99) = 0.13, *p* = 0.87), nor were significant differences observed between boys and girls (*F*(1, 100) = 0.16, *p* = 0.68).

### 5.3. Relationship between Attention Session 1 and Help Needed Session 2

There were no differences between the groups in the amount of help needed.

Among all children, there was a statistically negative correlation between sustained attention and the amount of help needed. This was similar to previous studies on third-party attention [12,20,21] *r*(102) = −0.28, *p* = 0.003.

When looking at the correlations within each of the cultural groups, the only statistically significant negative correlation between sustained attention and help needed was for the non-Mapuche group (*r*(38) = −0.42, *p* = 0.009), and the correlation was in the same direction as in previous studies [12,20,21].

When looking at the correlations by gender, there was a statistically significant negative correlation for girls (*r*(58) = −0.31, *p* = 0.01) but not for boys (*r*(44) = −0.26, *p* = 0.08).

When disaggregating the analysis, differentiating the children according to the toy whose construction they observed, Table 2 shows that those who observed the mouse needed less help from Toy Lady for the construction (*r*(51) = −0.36, *p* = 0.009); this may have happened because part of the information about what to do was contained in the mouse materials, as opposed to the difficulty of inferring a step in the construction of the origami frog [20,21]. Only in the group of non-Mapuche children was there a statistically significant inverse correlation (*r*(19) = −0.52, *p* = 0.02) between sustained attention and help needed; in the other two groups, there was no significant correlation between these variables (rural Mapuche children, *r*(19) = −0.33, *p* = 0.16; urban Mapuche, *r*(13) = −0.28, *p* = 0.34)

In the total sample, the group of boys who constructed the mouse did not present a significant correlation in these variables (*r*(21) = −0.36, *p* = 0.10), but the group of girls did (*r*(30) = −0.36, *p* = 0.04).

As for the boys who observed the frog making, the correlation between sustained attention in the first session and help needed in the second session was not statistically significant (*r*(51) = −0.13, *p* = 0.34). This result was similar for the three groups of children who observed the frog and unexpectedly had to build this toy: rural Mapuche, *r*(19) = 0.03, *p* = 0.90; urban Mapuche, *r*(13) = 0.11, *p* = 0.71; and non-Mapuche, *r*(19) = −0.41, *p* = 0.08. Both boys and girls did not show a significant correlation between these variables (*r*(23) = −0.04, *p* = 0.84, *r*(28) = −0.20, *p* = 0.28).

## 6. Discussion

The results obtained when comparing the three cultural groups show that rural Mapuche children spend more time in third-party attention in activities in which they are not directly involved. It seems that the practices of rural Mapuche families encourage third-party attention, much more so than in non-Mapuche and urban Mapuche families. Mapuche children would be more accustomed to learning from observation and collaborative participation in diverse family collective activities [28,44,45,46]. This finding is consistent with the study by Correa-Chávez and Rogoff [20], which concludes that indigenous children, from families whose practices are not as permeated by formal schooling, grow up in a community where they are expected to attend to events that occur in their environment and pay attention to situations in which they are not directly involved [13]; even when children do not participate in the productive activities of their caregivers, they can still monitor at times when their presence is needed [19,22].

Therefore, the development of third-party attention as a skill arises in constellations of cultural practices, which involve children’s living conditions (such as parents’ formal schooling experience, type of work and daily activities) and differently guide their development [47,48,49].

In rural Mapuche communities, children’s learning is based on observing and listening attentively and as a collaboration between children, youth, adults and elders in family activities [28,50,51]. The fact that, in Mapuche families, the mother is permanently at home and the father is engaged in agricultural work on his own land allows them to spend more time with their children, and, consequently, they develop a dense network of collaborative practices that makes it possible to pay third-party attention [27,28,33]. In this sense, participation in Mapuche spiritual ceremonies is highlighted, which requires the development of third-party attention and surrounding events, as occurs in other indigenous communities in the Americas [9,14].

On the other hand, urban Mapuche children showed less sustained attention to the activities of third parties, possibly because daily school practices involve few collaborative activities and demand a selective and focused type of attention [52,53]. Urban Mapuche families have more years of formal schooling than rural Mapuche families, and, therefore, it is possible that practices are encouraged to respond to the demands of the school system, thus changing the cultural patterns of attention and the learning of children [53,54]. These approaches are consistent with those of Coppens et al. [17], who conclude that indigenous immigrant families to the city engage less in traditional cultural practices because of increased formal schooling; as a consequence, third-party attention changes to be more direct and selective.

In the case of the non-Mapuche children, their caregivers had more than 12 years of formal schooling. Their work activities were carried out outside the home, which would limit the opportunity for these children to collaborate in activities with their caregivers [15]. Certainly, for them, school activities are an important part of the daily routine, with few opportunities to develop third-party attention in collective practices [13,55,56].

By contrast, the rural Mapuche children presented a lower amount of casual glance compared to the two other groups of children, probably because they were in sustained attention and inattention for the duration of the toy construction demonstration.

It is worth mentioning that, in this study, child inattention was present for a greater amount of time than in previous studies that evaluated third-party attention in sibling pairs from indigenous Mayan or Mesoamerican indigenous heritage immigrant families to the U.S. [12,20,21]. Perhaps in this study, not being siblings but friends, the children were not as “motivated” to see what their classmate was doing. It is inevitable that measurements in different cultural groups present different patterns, since they are themselves cultural devices that operate differently according to children’s skills [57]. The third-party attention task used in other studies was developed for children with Mesoamerican indigenous heritage from families with little formal schooling experience, as opposed to the participants in this study who came from families with more years of formal schooling [20]. It is important to view this divergence from the cultural differences between groups [40]. No group is less disadvantaged or deprived, but they simply develop their skills differently according to the opportunities, characteristics and needs of their families [58].

As expected, there was a significant inverse correlation between sustained attention to the construction activity and the help needed to assemble the toy, which is consistent with findings from other studies [20,21]. Learning occurred through simultaneous attention to multiple foci, without external monitoring or direction [11]. The greater the sustained attention, the less help the children needed to assemble their toy in the second session [21]. In this sense, the observation and listening involved in sustained attention developed learning in the children [24]. Specifically, this correlation was significant only for the non-Mapuche group, which is similar to the findings of Correa-Chávez y Rogoff [20] concerning Euro-American children. A limitation of this study was its sample size. This may have led to insufficient variability in the data so that the correlation between sustained attention to the toy-building activity and help requested by rural Mapuche children was not statistically significant.

The fact that this relationship between sustained attention and help was not significant for the urban and rural Mapuche groups is consistent with the findings of López et al. [12] in Mexican immigrant children with indigenous heritage, although the correlation was in the same direction (negative) as in previous studies. This may indicate the existence of other patterns of attention (not only sustained attention) for learning in Mapuche children, i.e., other ways of learning from observation and participation in community activities that could not be covered in this unit of analysis.

It is possible that rural Mapuche children do not learn exclusively through third-party attention. Perhaps they develop divided attention in order to be attentive to everything that happens around them, like “hummingbird attention”, whose attentional focus is concentrated on several events in a very fluid and skillful fashion, managing several ongoing events at once; for example, they could observe adults feeding chickens and collecting eggs [8,16]. Maybe rural Mapuche children develop divided attention in order to observe and participate in several activities without any of them being interrupted and thus learn to collaborate in their community [22]. In other to corroborate this supposition, it would be necessary to carry out a new study, with a microanalysis of attention, not only of five-second segments, because perhaps the casual vision encoded in each five-second segment could be divided attention in a frame-by-frame analysis. For this reason, we believe that it is necessary to use other ways of examining video recordings to analyze interactions during the toy-building activity and find patterns of attention in situ.

While previous research has suggested that practices associated with formal schooling are often substituted for other cultural practices among people who have extensive experience in formal schooling [4,15], children can be thought of as having more than one learning style by observation, with forms operating depending on individual and collective success and the resources available for learning [12,24].

In summary, rural Mapuche children had more time in sustained attention than the other groups, and this attention seemed to help them learn about toy construction, which is consistent with studies on third-party attention [12,20,21].This study supports the idea of a peripheral observer approach to learning by attention to social events, which is common in indigenous communities in Central America, whose children participate in a range of activities in their family and group, learning by observation and contribution [23].

Furthermore, this study demonstrated that the selective attention characteristic of hegemonic school learning is not the only “valid” way of acquiring knowledge and developing skills [8]. The implication of these findings, along with many more studies, could guide the development of Chilean school protocols that appeal to curricular flexibility and the adaptation of teaching strategies, respectful of children’s third-party attention, in order to allow for culturally sensitive learning of socio-cognitive skills [20,55]. This would make it possible to maintain the cultural attention of Mapuche children in their families, because schools would respect the ability to learn by observing others for the benefit of all [12].

Cultural attention to ongoing events could manifest itself as broad, acute, open and permanent in an active process of both noticing what is happening around in order to develop cultural skills or notice the needs of others and collaborating in solidarity for their resolution [4,56].

## Figures and Tables

**Figure 1 behavsci-14-00689-f001:**
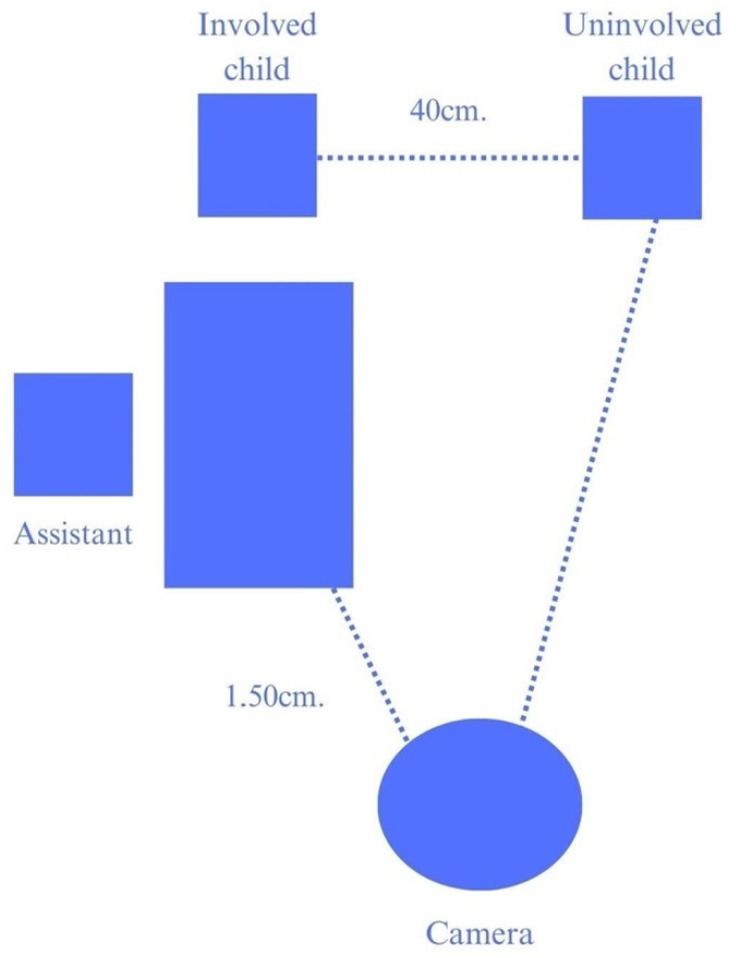
Schematic distribution of the top view of the furniture and place arrangement of the participants in the first session. Adapted from “Mexican-Heritage children’s attention and learning from interactions directed to others” by Silva et al. [21]. Reproduced with authorization.

**Table 1 behavsci-14-00689-t001:** Means and standard deviation, first session variables.

	Rural Mapuche (38)	Urban Mapuche (26)	Non-Mapuche (38)
Types of attention	*M (SD)*	*M (SD)*	*M (SD)*
All children			
Sustained attention	23.1 ^a^ (19.8)	14.4 ^a^ (11.4)	14.7 ^a^ (13.3)
Casual glance	4.2 ^b^ (4.3)	6.9 (6.4)	7.7 ^b^ (6.5)
Inattention	62.1 (21.5)	62.5 (21.5)	64.5 (22.1)
First uninvolved child (he or she observed the mouse construction)			
Sustained attention	26.8 (22.1)	17.0 (11.4)	15.7 (12.9)
Casual glance	4.4 (4.6)	8.0 (7.1)	7.8 (6.4)
Inattention	59.1 (22.6)	56.1 (22.1)	60.5 (23.0)
Second uninvolved child (he or she observed the frog construction)			
Sustained attention	19.4 (17.1)	11.8 (11.2)	13.6 (14.0)
Casual glance	4.0 (4.1)	5.8 (5.6)	7.5 (6.7)
Inattention	65.2 (20.5)	68.8 (19.8)	68.5 (21.0)

Note: In each row, superscript letters indicate statistically significant differences between cells with the same letter (according to the planned comparisons).

**Table 2 behavsci-14-00689-t002:** Mean (and standard deviation) of the help needed for toy construction and correlations with sustained attention in the first session.

Help/Correlation Score	Rural Mapuche	Urban Mapuche	Non Mapuche	All Children
First uninvolved child (he or she observed the mouse construction)				
Help needed *M (SD)*	24.4 (15.3)	21.9 (13.9)	27.6 (18.0)	25.0 (15.9)
Correlation (*r*) between sustained attention and help needed	−0.33	−0.28	−0.52 *	−0.36 *
Second uninvolved child (he or she observed the construction of the frog)				
Help needed *M (SD)*	51.0 (19.9)	56.9 (11.4)	52.8 (16.1)	53.2 (16.5)
Correlation (*r*) between sustained attention and help needed	0.03	0.11	−0.41	−0.13

* *p* < 0.05.

## Data Availability

The data supporting reported results, generated during the study, can be found on Laboratorio de Cognición y cultura, Doctorado en Psicología, Departamento de Psicología, Universidad de la Frontera, whose director is Paula Alonqueo Boudon.

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
