# Peer review of "Attention Paid by Children of Rural Mapuche, Urban Mapuche and Non-Indigenous Chilean Backgrounds to Interactions Directed at Others"

_behavsci, 2024, doi:10.3390/bs14080689_

Round 1
Reviewer 1 Report
Comments and Suggestions for Authors
Overall, there are several sentences throughout that are more wordy than needed and become run-on. Please edit for clarity, even if it means breaking sentences into multiple parts to make the reading smoother.
In addition, this study needs more emphasis in the conclusion, as well as the introduction, about why such analyses are important in a broader context – is it important to retain sustained attention that is being lost through schooling? Does the loss of sustained attention mean that children become more removed from their cultural heritage? Explain.
The Discussion section needs more explanation about the limitations of the study, including the subjectivity of the measures used, as well as any limitations in methodology, etc.
Introduction:
1) In the LOPI paragraph (paragraph 2) you say “they acquire the skills” – explain who you mean when you say “they”
2) Say more about how LOPI and attention to others are connected. It seems at the beginning of the introduction that you go back and forth between these topics instead of explaining how they are related.
3) In the sentence “opportunities to learn by attending events that are not directed to them are scarce,” give an example of what you mean by “events that are not directed to them” for Euro-American families.
4) Was “third-party attention” directly measured in the studies you describe, or merely assumed from the differences in children’s abilities to remember a construction task over time? Is there a chance that this observable behavior could be due to a different variable like having the cognitive schemas for construction and not necessarily “third-party attention”? How did they rule out other causes for the differences between children?
5) The paragraph that starts “unlike what happens in indigenous communities,” is worded in a very confusing way. Please rearrange the thoughts here to make it flow more clearly.
6) The explanation that “as children advance in schooling their selective attention becomes stronger” needs to be earlier in the introduction, because it gives context to why you’re talking about the amount of schooling.
Method:
1) Explain more about the exclusion criteria “belonging to a cultural group other than Mapuche or non-Mapuche.” Arguably that would include everyone. Do you mean “Mapuche or Euro-American”?
2) Explain more about how dyads were selected or assigned for the experiment.
3) In the paragraph about ethical safeguards, you explain that teachers were asked “information about the children.” What information did the teacher provide about the children?
4) Before the paragraph that starts “Sustained attention: intense observation,” you need a sentence or so to transition from the previous paragraph that introduces that you’re now discussing the codes that were used.
5) When explaining the codes, give some explanation of what determines the score on the scale from 0 to 69. Is it length of time engaged in sustained attention? Or something else? Do the same for “casual glance,” and “inattention.” The reader should be able to read how you’ve explained the coding, and then watch a video and have some idea of how to score the child they observe.
6) In the paragraph “help needed,” the list of numbers is inconsistent. For the first three, the authors put the number right next to its explanation, but then put “4 = Lady toy…” Please make this consistent across all of the numbers.
7) This reviewer is somewhat concerned that the amount of “help needed” is a very subjective assessment here, as the “Lady Toy” observes the child and when they need help she offers it. However, it seems like there is a lot of room for error here, where different kids may ask for help in different ways, some kids may process problems slower but not actually need help, etc.
Discussion:
1) Rewrite the paragraph: “Therefore, to develop of skills, such as third-party attention, arises in constellations of cultural practices that involve children's living conditions and differently guide their development (Bronfenbrenner, 1986; Mejía-Arauz, 2015; Worthman, 2010)” The wording here is too confusing to understand the message.
2) This reviewer is not understanding how participation in spiritual ceremonies requires the third party attention. Please explain further.
3) For the paragraphs that begin “In the case of the non-Mapuche children,” and “On the other hand, the rural Mapuche children,” it’s not clear how these paragraphs add to the discussion. Either integrate them more effectively or consider omitting them.
4) The explanation of the higher inattention scores as a sign of respect for adults (i.e., a Mapuche value) does not make sense in the context of the larger study, because the different groups did not differ in the amount of inattention, which we would expect if it was reflecting a Mapuche value, because not all of the participants were Mapuche.
5) The explanation of the significant inverse correlation between sustained attention and help needed is confusing. The fact that it was only significant in non-Mapuche children contradicts earlier studies, correct? I see that the authors have included studies that found the same thing. They also need to include the studies that contradict their findings and explain why they found something different than previous studies.
6) Regarding the Mapuche children not having a significant correlation between third party attention and need for help, is it possible there is not enough variability in the data? If Mapuche children did not need much help at all, there may not be enough variability to find significant differences/correlations.
Comments on the Quality of English Language
Some sentences in particular require rewording for clarity. Specifics are included in the overall comments for the article.
Author Response
Thank you very much for taking the time to review this manuscript. Please find the detailed responses below and the corresponding revisions/corrections to track changes in the re-submitted file. Please see the attachment.

Reviewer 2 Report
Comments and Suggestions for Authors
The reviewed study examines cultural differences in attention to interactions directed at others (third-party attention) by 9-11-year-old children of three cultural backgrounds in Chile: Rural Mapuche, Urban Mapuche and non-Indigenous. Third-party attention is a method of observational learning which is part of a constellation of features that constitute the theoretical framework of Learning by Observing and Pitching to family and community endeavors (LOPI; Rogoff, 2014). LOPI is a way of organizing learning in many Indigenous and Indigenous-heritage communities of the Americas. Implementing a two-session toy-construction task, and confirming expectations in line with previous studies, rural Mapuche children significantly attended more to the interaction not direct toward them compared to counterparts from the other two groups (session 1). Although significant differences were not found in amount of help needed in session 2 to construct the toy they had observed being constructed a week before, there was a negative correlation found between sustained attention in session one and help needed in session two for the first uninvolved child. Within cultural groups, this relationship was only statistically significant for non-Mapuche children.
This study is a great addition to the work on third party attention by including additional groups and highlighting specific practices of those groups. My suggestions include:
· - Consider capitalizing “Indigenous” throughout
· - For readers who are not as familiar with this line of work, it may be useful to more explicitly state how this study differs from the previous and what it adds.
· - In the beginning of the intro, there is some inconsistency about how “attention to interactions directed to others” is referred to. It might be useful to begin by describing this type of observation (which the study already does) and to state how it will be referred to for the rest of the study.
· -On p. 2 when describing third-party attention and changes with western schooling, it might be useful to also mention that third-party attention is also discouraged and may be punished in some school setting (if that’s also the case in Chilean schools).
· -It may be useful to use sub-headings in the introduction, in particular when presenting information on the current study which begins on p. 2, line 91.
· -On p. 3, it is stated that “Mapuche cultural practices are not so permeated by school knowledge,” but it might be clearer to say that they are “not so permeated by school-like practices” instead (line 99).
· -When stating the hypotheses on p. 3, the language could be more direct. It is currently stated that “it could be assumed” (line 132) and that “one might expect” (line 135), when it could be more explicitly stated.
· -Further subheadings would be useful when discussing the participants, and the demographics would also be more useful if they were broken up by cultural group. Although gender analyses are mentioned in the results section, I don’t think a gender breakdown is included in the participants section which would also be informing.
· -More information is needed on who the Toy Lady is (p. 4). What is her background? What language(s) does she speak? Did the same Toy Lady guide the activity for all three cultural groups? Could the Toy Lady’s background play a role in the children’s behavior?
· -On p. 5, line 207, the Toy Lady is referred to as “the research assistant” again after being introduced as the Toy Lady on the previous page. The same term should be used for consistency.
o The “research assistant” is referred to again on p. 6, line 247. Is it the Toy Lady? Also, did the research assistant/Toy Lady ask the children to sign the assent form when they were about to construct the toys or was this in a previous occasion?
· -It might be easier to use “they/them” instead of “his or her,” for example, p. 6 lines 263-264.
· -When describing the coding scheme, the minimum and maximum scores need a little elaboration. For example, is the maximum score for sustained attention 69 because the child with the most sustained attention had that code for 69 different 5-second segments?
· -Table 1 on pp. 7-8 was a little confusing at first because the heading “First uninvolved child” is on p. 7 when it describes the first set one numbers on p. 8.
· -More description is needed on what the children were doing when they were not attending to the third-party interaction, especially since most of the activity was coded as “inattention” across all three groups. For example, were they just playing with the “do-nothing machine”? Did the children at any point try to interrupt the ongoing activity? Were there any differences in the type of inattention across the three groups?
· -It may be my mistake, but the correlation values described on p. 9 (lines 363 and 365) don’t seem to match up with the values in table 2. The degrees of freedom also seem to reflect the sample size rather than n-2.
· -Throughout the study, it might help to refer to “schooling” as “formal schooling” (e.g. p. 9, line 399).
· -The paragraph on p. 11 that begins on line 474 could be clarified and expanded on a little more.
· -The direct implications that the findings have for schools and families also need to be more explicitly stated at the end.
Comments on the Quality of English LanguageThe language used in the study was easy to follow. Minor editing is suggested:
· -The use of “Lady Toy” is a bit confusing and perhaps “Toy Lady” should be used.
· -P. 1: In “they acquire the skills” (line 29), it’s a little unclear who “they” are, but it might be best to say, “children acquire the skills…”
· -P. 2: Might be best to use “extensive schooling” instead of “high schooling” at least in the beginning (line 46) and then a short-hand term can be used if needed.
· -P. 2: Might be better to say “…participate in family and community helping activities” (line 47)
· -P. 2: A “to” is needed for “opportunities to learn by attending to events that…” (line 51)
· -P. 2: An “and” is needed for the in-text citation, “Correa-Chávez and Rogoff” (line 54)
· -P. 2: “limited-schooling” might work better than “low-schooling families” (line 55)
· -P. 2: the wording, “required to make the toy” (line 60) makes it seem a little forced when the children were simply invited to make the toy they had previously observed.
· -P. 2: “Euro-American sibling groups and the Kaxlaan Mayan sibling groups, the attention was less…” (lines 63-64). An “s” is needed to make group plural, but “sibling pairs” can also be used.
· -P. 4: It would be clearer to say “Purposive sampling was used based on the following…” (line 151).
· -P. 4: Since the population of interest is described on the previous page, it might be good to explicitly state that the sample consisted of 102 children….(line 157)
· -P. 4: Rewording is needed when describing the task on line 183. For example, it might be clearer to say that a research assistant instructs each child how to make a toy, one at a time, taking turns.
· -P. 5, line 227: The “child’s making” is unclear. I believe it was meant to read that her task was to “supervise the child’s construction of the toy”.
· -P. 6, lines 238-239: Might be clearer to say, “the child took their mouse and frog with them”
· -P. 6, line 244: unclear what “authorization for the application” refers to. Maybe the document described in the prior sentence?
· -P. 6, lines 247-248, the description of the development of the schedule is a little unclear. Was it meant that the schedule for children’s participation was developed alongside the teachers?
· -P. 9, line 404: perhaps should read “the development of skills” instead of “to develop of skills”
· -P. 11, line 477: probably needs to read, “but this possibility…” instead of “but his possibility”
Author Response

(The authors gave the same response as above.)

Reviewer 3 Report
Comments and Suggestions for Authors
This study examined third-party attention among children Mapuche and non-Mapuche children in Chile, replicating previous work among Guatemalan Maya children and children from Mexican and Central American immigrant families in California. Consistent with previous work, the authors found more sustained third-party attention among the Mapuche children and found that the children learned by attending to others’ activities. However, the findings differed from previous studies in that the Mapuche children engaged in less third-party attention compared to what has been found in the other populations. The study is interesting, relevant, and makes an important contribution to the literature. The following revisions divided into three categories below (theoretical, methodological/results, and English language) would improve the manuscript.
Theoretical
Throughout the manuscript there seemed to be a conflation of sustained attention and third-party attention. The two concepts are related within the umbrella of Learning by Observing and Pitching In (LOPI), but they are not the same thing. For example, in the article’s second paragraph starting in line 31 the definition of third party attention is wrong. The definition given (maintaining two objects or activities in focus) is closer to the definition of simultaneous attention (Rogoff et al, 1993; Chavajay & Rogoff, 1999; Correa-Chavez et al., 2005) than to third-party attention. In third-party attention it is not necessary to keep two or more foci of attention – what matters is that the attention is focused on something directed at others (not oneself). There is an argument to be made about the relationship between third-party attention and simultaneous attention – indeed there is a mention of this in the discussion (more on that below), but here the definition is wrong and should not be included as it is not the focus of the study. (Additionally, as the first study of third party attention Correa-Chávez & Rogoff, 2009 should be cited here when third party-attention is first mentioned on page 2.) Similarly on page 2 (lines 79-82) there is more discussion of simultaneous attention – this should also be moved to the discussion (again more on this below) because it is not relevant to third-party attention.
Methodological/Results
We need more information about the toy lady. Was she a member of the community? A teacher? Were any of the sessions conducted in Mapudungun? Were the children comfortable with her or were they more formal? Also, where was the study carried out? In school during school hours? Or after school? Any of these may have indicated to the children that the task was a more formal school activity and may have indicated to the children that they should be behaving in ways consistent with school.
In the coding section, the last sentence in each paragraph includes scores, and it is unclear what these refer to. Is this the range for each coded category? I suspect this might be the case since there are no ranges in the reported in the results. If so, it should go in the results section and not here. If these are not ranges, and there is a reason for them in the coding section, this should be explained.
In the results, it looks as if raw numbers are used instead of proportions. If raw numbers are used the ranges for each of the coded categories should be included
The three paragraphs on page 9 were confusing in that they appeared to restate the same thing. Only after reading them through over a few times did I understand that the information was not repeating. Perhaps headers could make these differences more clear. I believe these are the findings:
- There were no differences between the groups in the amount of help needed. (This was not included in the paper and only inferred from the table. It should be included in the results.)
- Among all the children there was a negative correlation (significant) between sustained attention and amount of help needed
- When looking at the correlations within each of the cultural groups, the only significant negative correlation between time 1 and time 2 was for the non-Mapuche
- When looking at the correlations by gender, there was a significant negative correlation for girls and not for boys
These pattern of results makes me wonder if casual attention was coded differently in this study compared to previous studies. In previous studies the amount of time (1 second or 2 or more) was not as important as the perceived interest of the child when they looked. The description provided here in terms of time – along with the inclusion of issues related to simultaneous attention makes me think that these casual glances were not casual, but purposeful. The original coding for the third-party attention studies included both purposeful and casual glances, but purposeful glances were extremely rare in the study and thus not included in the analysis. However, in this study they may be more prominent, especially if as you suggest the children were less willing to look over because they were told to wait and do something else.
Were purposeful glances coded? Can they be differentiated from casual glances? If they can, then these too would be paying attention to the actions of others and could be included with sustained attention. (This may or may not alter the pattern of results.) If purposeful glances were not coded, then this could be discussed further in the discussion in several ways which I will leave up to you to decide upon. For example, it could be a limitation, or a future direction in terms of examining cultural variability.
Lastly, the differences between this and previous studies is a contribution from this study that should be highlighted more in the discussion. If your examination leads you to truly believe that this is a different cultural pattern (rather than the children behaving in a “school like way” because they were in school, during school hours, and with a teacher), then this should be explored more fully, and highlighted with its own heading. Here, is where the discussion of simultaneous attention makes more sense, especially if you suspect that the pattern of paying attention to others’ activities looks more like simultaneous than a sustained attention. In both cases, the overall pattern may be third party attention, but it may look different in these two distinct Indigenous cultural contexts, even though in both communities children are included in productive adult activity. If this is the case, then this should be developed more – if this is true, then why would it look different with sustained looking in one and perhaps quick purposeful glances in the other?
Devote more time here and elaborate the idea – I think this would be an excellent contribution.
English Language
· Throughout the manuscript “Lady Toy” should be changed to “Toy Lady” as Lady Toy implies that “Toy” is the lady’s name. Toy Lady indicates that this is the lady that makes the toys
· In abstract: Third-party interaction (line 10) should read third-party attention
· Line 27 Learning by observing and pitching in to (not pitching in in)
· Page 2 line 54 “y” should be “and”
· Page 2 paragraph starting on line 65 has an awkward structure and should be reworded or immigrant to the U.S. put in parentheses to improve flow.
· Line 77 remove institutions from “school institutions” or reword to “the institution of school”
· Page 2 line 96 “They preferably” - does this mean the community? This phrasing is unclear, who prefers? Also preferably isn’t the best word here, consider: “The community has a an agricultural-livestock…”
· Page 3 change “educational offer” to “educational offerings”
· Page 6 line 276 “he” should be either “they” or “he or she”
· Page 6 line 276 sidelong glance should be removed, as it can be a sustained sidelong glance – that does not imply a quick uninterested glance – in fact often it is the opposite may be inferred
· Help needed section (pg 7): change to past tense in whole section (“is” to “was”)
· “Analysis of information” change to “Data Analysis”
· Are a posteriori tests post-hoc tests? If so, “post-hoc” should be used
· Page 8 line 336 change “obtained” to “found”
· Pg. 9 line 361 “It was expected to obtain a negative…” should be changed to “We expected a negative …
· Page 14 line 392 “sustainedly” not sustainably
There likely are other English language slips peppered in the manuscript, these were the ones I saw, but a good proofreader is recommended.
Comments on the Quality of English LanguagePlease review the manuscript thoroughly, while I indicated multiple places where the English could be improved, it is also likely I missed other places.
Author Response

(The authors gave the same response as above.)
